# The Plant-Like Structure of Lance Sea Urchin Spines as Biomimetic Concept Generator for Freeze-Casted Structural Graded Ceramics

**DOI:** 10.3390/biomimetics6020036

**Published:** 2021-05-31

**Authors:** Katharina Klang, Klaus G. Nickel

**Affiliations:** 1Institute of Glass and Ceramics, Friedrich-Alexander-Universität Erlangen-Nürnberg, Martensstraße 5, D-91058 Erlangen, Germany; 2Department of Geosciences, Applied Mineralogy, Eberhard Karls Universität Tübingen, Wilhelmstraße 56, D-72074 Tübingen, Germany

**Keywords:** structural graded material, freeze-casting, biomimetic, microcomputed tomography (µCT), porous ceramic, sea urchin spine, alumina, fracture behavior

## Abstract

The spine of the lance sea urchin (*Phyllacanthus imperialis*) is an unusual plant-akin hierarchical lightweight construction with several gradation features: a basic core–shell structure is modified in terms of porosities, pore orientation and pore size, forming superstructures. Differing local strength and energy consumption features create a biomimetic potential for the construction of porous ceramics with predetermined breaking points and adaptable behavior in compression overload. We present a new detailed structural and failure analysis of those spines and demonstrate that it is possible to include at least a limited number of those features in an abstracted way in ceramics, manufactured by freeze-casting. This possibility is shown to come from a modified mold design and optimized suspensions.

## 1. Introduction

Many biological tissues and devices have been systematically studied due to their remarkable engineering properties e.g., the toughness of the coconut endocarp [1,2] or the fire resistance of particular barks [3] with the objective of solving technical problems. In recent years, more and more of these natural materials have been systematically studied with the objective of solving technical problems using abstraction, transfer and application of knowledge gained from biological models [4,5,6]. Several plants, animals or specific organs of these organisms have evolved strategies to cope with sudden potentially destructive mechanical loads and impacts caused by falling, rock falls, attack of feeding animals and various other environmental loads such as wind gusts or wave movements. The typical mechanism of some biological structures to survive overcritical loads is to dissipate energy by allowing the failure of substructures, whose loss is not endangering the function of the whole organism. 

Among the structures to realize simultaneously the properties of structural strength, light weight and fluid transport many plants have developed stems and stalks. Many stems have typical cross-sections [7,8], which show below an epidermis concentric cellular structures with different properties, such as a cortex, a phloem, a xylem and a pith. As an abstracted form this can be viewed as a partially or fully filled tube, hence displaying a core–shell structure with several possible differentiations of the substructures.

While this is commonly found in the plant kingdom, where the cellular construction of stems from organic matter allows many biological functions, we do not associate these structures with brittle natural materials. However, the structure of sea urchin spines, although being part of an animal, is remarkably close to plant stems. In this paper we focus on the aboral spines of *Phyllacanthus imperialis.* A cross section from such a spine is shown in Figure 1 to illustrate the plant-alike structure on a micrometer level. Three different cellular concentric structures are characteristic for the spine in total differentiating in their arrangement and density within the microstructure. The building material of the three-dimensional skeletal meshwork is high-magnesium calcite (Ca_×_Mg_1−×_CO_3_) and is known as stereom [9,10]. Investigations have shown that the MgCO_3_ content varies between 2 to 12 mole percent in the stereom structures of the spine. The stereom structure in the center of the spine is the medulla, which is characterized by a highly permeable cellular structure. Denser cellular stereom structures enclose the medulla being defined as radiating layer. The radiating layer is encircled by an almost dense shell—the cortex. The pores of the stereom are filled with stroma consisting of fluids and several organic compounds such as extracellular fibrils and connective tissue cells [11].

On the nanometer, scale the stereom meshwork consists of highly ordered polycrystalline mesocrystals [12]. These mesocrystals are characterized by highly ordered nanoparticle-sized domains, oriented parallel to the *z*-axis. Small quantities of organic macromolecules (<1 wt.%) are within the magnesium calcite domains [13]. The hierarchical organized lightweight construction of the spine is characterized by remarkable mechanical properties such as high strength [14,15] and beneficial failure behavior dissipating energy in large quantities [16] despite their cellular, brittle microstructure. The transfer of such desirable properties such as strength, high energy dissipation capacity and lightweight into ceramic-based materials and structures is of particular interest for the area of separation technology/filtration (e.g., particle filtration, liquid filtration), in chemical and thermal process engineering (e.g., catalyst carriers, pore burners), medical technology (e.g., bone substitute) and in lightweight construction under high pressure. 

It has been shown in [17] that the manufacturing of a biomimetic porous ceramic containing cap-shaped layers of alternating porosities is basically feasible utilizing starch-blended slip casting. This modified slip casting method has its limitations. Ceramics manufactured with this method lack the homogenous and ordered pore geometry. Other techniques like the layer-by-layer deposition [18,19], filtration [20,21], hydrogel casting [22] and electrophoretic deposition [23] are time-consuming and size limiting processes, which enable to fabricate ceramic layers with a thickness of approximately 200 μm only. Freeze-casting produce, in contrast to the techniques mentioned above, porous ceramics with complex, three-dimensional pore structures that can also contain at least two different levels of open porosity and an ordered pore alignment. Mimicking those features is required in order to manufacture the abstracted concepts of the spine of *Phyllacanthus imperialis* into a porous bioinspired ceramic. In general, the freeze-casting technique has clearly become a focus of attention in the last decade concerning the manufacturing of complex pore geometries and open interconnected pore systems due its simple operation, cost efficiency and environmental friendliness.

The manufacturing process uses a ceramic suspension, which is directional frozen and then sublimated before sintering (Figure 2). This process provides materials with a unique porous architecture, where the porosity is almost a direct replica of the frozen ice crystals. 

The pore morphology, orientation and average pore size can be tailored by altering process parameters such as slurry concentrations [24,25,26], freezing temperatures [25,27,28,29], cooling rates [25,30,31,32] and use of additives [25,26,33]. It is important that the prepared slurries are stable during the entire duration of the freezing stage. Since the solvent initially present in the slurry is converted into solid, that is later eliminated to form the porosity in the ceramic, the pore content can be adjusted by tuning the slurry concentrations. The porosity of the ceramic is directly related to the volume of the solvent. A wide range of porosities, approximately from 25 to 90%, can be achieved via freeze-casting [25]. The total porosity is also depended on numerous additional parameters affecting the packing of particles between the solvent crystals such as the nature of the solvent, its viscosity, the particle morphologies and size distribution [26]. The solidification behavior of the freezing vehicles and thus the pore structure left by the frozen vehicles is affected by the freezing temperatures [25,27,28,29]. The pore channel size decreases significantly with lower freezing temperatures, regardless of any possible microstructure variations in the individual specimens. Porosity also decreases as freezing temperatures decline. With decreasing freezing temperatures, the solidification velocity is increased inducing smaller lamellar ice crystal spacing and thus pore channel size. This freedom adds significant difficulties when trying to understand the underlying principles that govern the relations between processing and the obtained microstructure. The pore structure of freeze-cast ceramics is determined by the morphology of the solidified fluid [34]. Water is the most common fluid for the freeze-casting process, because of its environmental friendliness, low cost, easy handling and best application chances. Furthermore, employing a unidirectional (=conventional) freeze-casting technique, lamellar pore structures can be fabricated. Generation of wall interconnectivity during the solidification can be attained by modifying the character of the suspension. This can be achieved in a water-based suspension by using additives. To suppress the preferential lamellar growth of ice crystals extensively, it is known from previous studies [24,35] that gelatin as additive in the conventional freeze-casting procedure induces a cellular pore morphology exhibiting an increased interconnectivity of the cell walls. The water molecules, surrounding the gelatin, form a discontinuous network and prevent the concurrent preferential lamellar growth of ice crystals. Instead, spatially separated ice nuclei are formed in the gelatin network. Thus, the ice crystals are reduced in size and show a polygonal morphology [24].

Using the principle of the bottom-up approach of biomimetics [36], we analyze in-depth the lance sea urchin spine *Phyllancanthus imperialis* quantitatively in terms of its form-microstructure-function relationship and its mechanical properties to understand the operating principles of the relevant structures. These are interesting for various technical implementations, i.e., structurally graded ceramics with anisotropic pore morphologies being used as particulate filters for vehicular emission control. High-resolution micro-computed tomography (μCT) proved to be ideally suited to analyze the internal structures of the spine non-invasively regarding the strut configuration, pore sizes and porosities. Based on the results of the relation between the mechanical and microstructural properties, we present a new detailed structural analysis and failure observation of those spines and demonstrate the inclusion of a limited number of those abstracted features in freeze-casted ceramics. A modified mold design and optimized suspension was used in the freezing process to obtain the gradation features in the ceramics.

## 2. Experimental Procedure

### 2.1. Lance Sea Urchin Spines

The sea urchins were not killed for the investigation of the spine material. The investigated primary spine material belongs to the tropical cidaroid *Phyllacanthus imperialis* [37]. The spine material was purchased from a material supplier (Mineralien- und Fossilienhandlung Peter Gensel, Weimar, Germany).

### 2.2. Freeze-Casting

Unidirectional alumina scaffolds inspired by the spine’s microstructure were manufactured via the freeze-casting route from a water-based alumina suspension. Gelatin as additive was used in the water-based suspension. Water suspensions of gelatin powder (Ewald-Gelatin GmbH, Bad Sobernheim, Germany) were prepared with a solid loading of 6.8 and 3.5 vol.% (Table 1). Homogenization of the gelatin suspension was carried out at 60–70 °C by stirring for 1 h. Under continuous stirring of the gelatin suspension, the 99.99 % α-alumina powder (TM-DAR Taimicron of Taimei Chemicals Co, d50 = 1.2 μm, ρ = 3.98 g/cm^3^) was added slowly in small measures to obtain a solid load of 16.5 vol.% by dispersing them with 0.47 cm^3^ of Dolapix CE 64 (Zschimmer & Schwarz GmbH & Co KG Chemische Fabriken, Lahnstein, Germany), respectively (Table 1). The volume of Dolapix was fixed at 0.5% of the solid content (recommendation of the manufacturer). The dispersion was carried out with an ultrasonic processor UP400S (Hielscher Ultrasonics GmbH, Teltow), equipped with 20 mm titanium sonotrode (Amplitude 50%, cycle 0.5 s), for every alumina–gelatin suspension for 5 min. A constant rotation of the beaker at the titanium sonotrode ensured a homogeneous distribution of the alumina particles and temperature. The temperature of the alumina–gelatin suspension was monitored and kept below 80 °C to avoid hydrolysis of the gelatin.

Polytetrafluorethylene (PTFE) tubes (with an inner diameter of 25 mm and a height of 35 mm) were used as molds that were placed on a copper plate with a thickness of 3 mm (Figure 3A). Before filling the alumina suspension into the PTFE tubes, the copper plate was cooled at a temperature of −60 °C. After 4 h of directional freezing, the frozen specimen was carefully removed from the PTFE tubes by slightly heating the PTFE tubes (Figure 3A). The water-based ice crystals were removed via sublimation. The obtained green bodies were sintered in a conventional furnace. Each intermediate temperature was achieved with a constant ramp speed of 1 °C/min. In the first stage the temperature was increased to 550 °C and dwelled for 2 h; subsequently the specimens were heated to 900 °C and dwelled for 2 h. Finally, the target sintering temperature of 1350 °C was reached and held for 4 h.

The basic mold design was manipulated through the addition of silicon and copper plates (Figure 3B). To test the manipulated mold design, the slurry composition corresponding to the ceramic ‘Polygonal cells’ (see Table 1) has been used for this purpose. The manipulated mold design consists of triangle-shaped silicon slices being arranged in a circular array and alternating with the copper plate at the bottom of the mold (Figure 3B). Alternating copper stripes (thickness = 500 μm) are placed at the side walls.

### 2.3. Sample Preparation and Characterization

The spine segments were characterized using optical light microscopy and scanning electron microscopy (SEM). For determining the quantity of the medulla, radiating layer and cortex in the spine segments, only the digital light microscope (Hirox RH-2000, Hirox Europe Ltd., Limonest, France) was used. The dimensions (length, diameter) of the spine segments were determined with a digital caliper. The microstructure of uniaxial loaded core cylinder of the spine, however, was investigated utilizing the SEM. After the first significant acoustic signal in the initial stage of loading, the uniaxial compression was stopped to examine the cracks in the microstructure of the core cylinder. The porosity of the spine segments was determined gravimetrically by assuming a density, *ρ*, of 2.711 g/cm^3^ for calcite [38]. Therefore, the volume of the samples was calculated by measuring the area of the cross section, *A*, on the microscopic images and multiplying it with the mean height, *h*. Together with the mass, m, the porosity was estimated from the following equation:(1)$$\phi \left(\%\right)=\left(1-\left(\frac{m\left(g\right)}{A\left({\mathrm{cm}}^{2}\right)\cdot h\left(\mathrm{cm}\right)\cdot \rho \left(g\cdot {\mathrm{cm}}^{-3}\right)}\right)\right)\times 100$$

The microstructure of the fabricated ceramics was investigated using SEM. Pore sizes (pore axis lengths, pore diameters) were determined utilizing the software Fiji 2.0.0 (ImageJ). In total, 100 pores were considered for each ceramic species. The porosity of the ceramics was determined using Equation1). Instead of the density of calcite, a theoretical density of α-Al_2_0_3_ of 3.97 g/cm^3^ [39,40] was implemented in Equation (1).

### 2.4. Scanning Electron Microscopy (SEM)

Spine segments (cut parallel and perpendicular to the *z*-axis) and ceramic samples were investigated with a scanning electron microscope (Tabletop Microscope TM3030plus, Hitachi Ltd., Tokyo, Japan) at the Institute of Geosciences of the University of Tübingen. The backscatter electron mode (BSE) has been applied for the microscopic analysis at an acceleration voltage of 15 kV and at a working distance of 6500 μm.

### 2.5. Micro Computed Tomography (µCT) and Data Processing

The upper part of a spine (Figure 4, red box) was scanned in a nano CT scanner (phoenix nanotom m, Fa. GE Sensing & Inspection Technologies GmbH, Germany) at the Institute of Textile Technology and Process Engineering Denkendorf (ITV). The samples were scanned at 800 kV/180 μA with an exposure time of 800 ms to a resolution of 1.27 μm per voxel. The acquired two-dimensional X-ray images were reconstructed with a Filtered Back Projection reconstruction algorithm. The μCT data are analyzed and reconstructed into three-dimensional objects using the Fiji 2.0.0 (ImageJ) software package. In order to determine the porosity xy-images were binarized using the iterative ‘default’ threshold based on [41]. Based on this binarization process, the software calculates the amount of black and white in the image. The algorithm of the plugin BoneJ [42] has been used to determine the thickness of the struts. For this purpose, binarized images have been utilized for the evaluation of the thickness as well. The pore axis lengths were determined from the reconstructed three-dimensional objects applying the mode ‘fit ellipse’.

### 2.6. Mechanical Testing

Uniaxial compression tests were conducted in a universal testing machine (Instron 3180, Instron Deutschland GmbH, Pfungsstadt, Germany). The spine segments and ceramic cylinders were placed on a Si_3_N_4_ plate and pressed against a tungsten carbide compression die. The force was measured simultaneously by a force transducer. A photograph and schematic view of the measuring principles of the universal testing machine is illustrated in [16]. All experiments were performed with a crosshead movement speed of 0.5 mm/min. During the uniaxial compression tests of the specimen, the fracture behavior and the displacement was monitored by a video extensometer (LIMESS RTSS-C02, LIMESS Meßtechnik GmbH, Krefeld, Germany).

From the geometry of the spine specimens, the maximum compressive strength has been calculated from the load at the first crack formation. The displacement recorded from the video extensometer was used for calculating the Young’s modulus from the slope of the linear elastic increase in the stress–strain plots.

## 3. Results

### 3.1. Sea Urchin Spine as Concept Generator

The microstructure of the adult, aboral spine of *Phyllancanthus imperialis* is composed of three different concentric structures being referred as medulla, radiating layer and cortex from the inside to the outside (Figure 5A). The cortex is a dense, but permeable shell that differentiate essentially from the cellular stereom arrangement of the medulla and radiating layer. A stacked µCT section of the cortex reveals several pore channels within the microstructure being oriented perpendicular to the *z*-axis (Figure 5B). Besides the small pore channels, the external morphology of the cortex contains also comparably large pores appearing very irregularly in their shape. Their pore channels, however, cannot be pursuit deeply into the internal microstructure of the cortex and disappear in the outer half of the cortex (Figure 5B). The quantity of the pores in the microstructure of the cortex increases visually from the outside to the inside, which is associated with an increase of the porosity from the outside to the innermost cortex structure: from an average of 11 ± 0.9%, the porosity increases up to an average of 18 ± 1.0% (Figure 6A). In total the cortex is characterized by an average porosity of 15.2 ± 3.2% (Figure 6A). The external surface of the cortex is characterized by an average pore size of 28 ± 10 µm (Figure 6B). Smaller pore sizes can be found in interior structures of the cortex: the average pore size is constant within the interior and varies between 15 ± 8 µm and 17± 8 µm (Figure 6B). 

The µCT reconstructions and sections visualize the inner mesh structure of the radiating layer and medulla (Figure 7). The radiating layer composes of thin sheets of stereom layers being separated by a regular arrangement of cross struts. The separation distance between the sheets appears regular and approximately equal to the thickness of the individual sheets. Each individual sheet is permeated by oval-shaped channels appearing in their multitude very even. Each sheet comprises a strict arrangement of pore rows parallel to the *z*-axis. The rows of pores are, however, arranged with a minimal misalignment along the *z*-axis meaning that every second pore row is congruent. The cross struts connecting each individual stereom layer are orientated perpendicular to the stereom sheets and are arranged in a periodic way. Every second row of cross struts is congruent analogously to the arrangement of pores. An analysis of the pore axis lengths (Figure 8A), a_1_ and a_2_, of the pores on the zx- and zx-plane (Figure 7) indicates similar average values (zx-plane: a_1_= 17.7 μm, a_2_ = 10.1 μm; zy-plane: a_1_ = 16.8 μm, a_2_ = 9.5 μm). 

The spine microstructure is characterized by a columnar arrangement of the stereom mesh (Figure 7). The stereom mesh consists of several long trabecular rods, which are linked by cross struts in such way to produce irregular polygonal passages parallel to the *z*-axis. The number of trabecular rods forming an individual columnar stereom varies from 4–8. The lateral pores on the zy-section are regularly formed and arranged in an offset manner to the adjacent pores. The average pore axis lengths of the polygonal passages (a_1_ = 22.0 μm, a_2_ = 17.4 μm) on the xy-section and of the pore rows on the *zy*-axis (a_1_ = 19.6 μm, a_2_ = 14.7 μm) are larger compared to the pores from the layered stereom type (Figure 8A). The smaller difference between the average pore axis lengths of the columnar rows indicates that the pores have a rather circular than an oval character. 

The average porosities of the three structural units of the spine are demonstrated in Figure 8B. The stereom mesh in the center of the spine, the medulla, is characterized by an average porosity of 85.7 ± 2.8%. The dense, regular arrangement of the stereom sheets in the radiating layer has an average porosity of 76.6 ± 3.0%. In comparison, the cortex is the densest of the three structural units and was characterized by a porosity of 15.2 ± 3.2%.

The calculated strut thickness distribution of the microstructure indicates that the stereom sheets from the radiating layer appearing as lateral strut routes (on the xy-plane) and the polygonal passages from the medulla are clearly separated from each other (Figure 9A). Thus, there is no transition zone regarding the strut thickness between both structural types. There is a clear border between the polygonal passages and the lateral strut routes: the polygonal passages are characterized by a strut thickness of <10 µm, whereas the stereom sheets have a thickness of >10 µm (Figure 9A,B). Based on the µCT data, the average strut thickness for the stereom structures in the medulla and radiating layer has been determined to be 8 ± 1.6 µm and 13.7 ± 2.0 µm, respectively. 

The mutual distribution of the two stereom structures in spine microstructure forms a superordinate structure, which is here described as a superstructure (Figure 9A). In this case, the strut configuration is slightly modified resulting in a subordinate structure, which is not obviously visible on every cross-sectional level. This kind of superstructure can be limited to a particular cross-sectional level and is therefore visible on this level. Taking the stereom structures of the radiating layer and medulla together, a superstructure becomes visible on the xy-plane (Figure 9A). Each individual stereom sheet is linked with the cortex microstructure. Starting from the cortex, each stereom sheet runs directly to the nearest side branch of the medulla in a straight to slightly curved manner. Consequently, a group of several stereom sheets forms a ‘wedge-like’ structure that encloses a side branch of the medulla. This sort of subordinate structure, where a group of stereom sheets encloses a side branch of the medulla, and is here called a ‘wedge’ (Figure 9A). In total, 13 wedges have been determined in the adult, aboral spine of *Phyllacanthus imperialis*.

To quantify the structural relationship of the spine segments with the mechanical behavior in uniaxial compression, the spine microstructure was divided into a shell (=cortex) and compliant core (=radiating layer and medulla). A simplification of the spine microstructure into a core–shell structure helps to model the influence and behavior of the cortex in the spine segment during uniaxial compression. A ‘structural factor’ was, therefore, created to summarize the quantity of the cortex in the spine segments (Figure 10A). The structural factor is composed of the radius of the spine segment, a, which was normalized by the cortex thickness, t (Figure 10A: marked in red). Such a structural parameter, expressed as radius to thickness ratio a/t, was first established and utilized by [43,44,45], who used the a/t ratio to characterize and model the shell-core system of plant stems, animal quills and bird feather rachis in terms of their mechanical efficiency. The a/t ratios of the structural elements in cross section (xy-plane) over the entire spine length is summarized in Figure 10B (= black squares). The a/t ratio increases from the spine tip to the base. Low a/t ratios are, in principle, an indication for a large cortex thickness, which is clearly the case for the spine tip. A large proportion of the cortex is, therefore, associated with low values of the porosity in the spine segment (Figure 10B, red circles). With a decrease of the cortex thickness, the porosity increases constantly up to a value of approximately 67% (Figure 10B, red circles).

The dependency of the Young’s modulus and maximum compressive strength on the microstructural core–shell design and accompanied porosities of the spine segment is displayed in Figure 11. The color bar indicates the a/t ratio of the measured, uniaxial loaded spine segments. The maximum compressive strength (Figure 11A) and Young’s modulus (Figure 11B) is negatively correlated with the porosity of the spine segments. It means in relation to the core–shell design that a decrease of the a/t ratio in the microstructure is associated with an increase of the Young’s modulus and maximum compressive strength. A decrease of the a/t ratio is accompanied with a rise of the volume fraction of the cortex. The volume fraction of the cortex influences also the fracture behavior of the spine segments. Two main failure modes have been identified, named failure mode I and II (Figure 11). Spine segments undergo failure mode I are characterized by a/t ratios of smaller than 9.2. Failure mode II occurs in spine segments being marked by large a/t ratios of more than 9.2. The change of the failure modes is in the porosity range of approximately 55% (Figure 11: light-yellow line). 

Typical stress-strain curves and corresponding microphotographs of the failure modes of the spine segments are shown in Figure 12. Failure mode I shows a high plateau strength after the maximum compressive strength, which is comparable to the failure behavior of cellular materials in general (Figure 12A). A sharp decrease of the stress after the maximum compressive strength followed by a constant low stress level are characteristic features of failure mode II (Figure 12B). The core–shell system reacts to the compressive stress, which dependents on the volume fraction of the cortex in spine segment. Failure mode I is characterized by a rigid connection of the cortex-core system, because the cortex remains attached to the core during the entire compressive loading. A breakdown of the core–shell system can be observed in failure mode II, since the cortex flakes off the spine structure after reaching the maximum compressive strength of the spine segment.

In detail the fracture behavior of failure mode I are as follows: after the subsequent linear elastic increase, the first drop in stress is accompanied by the generation of individual horizontal cracks, which occur in the lower region of the spine segment (Figure 12A,b). Originating from the horizontal ring crack, vertical cracks initiate to propagate through the spine segment (Figure 12A,c). The stress oscillates on a plateau level after crushing of the lower region between the bottom and ring crack by continued spallation. Simultaneously, a lath-like segment splits off from the spine structure, which is relatively small compared to the remaining spine segment (Figure 12A,d). A significant drop in the stress can be observed after complete spallation of the lath-like segment (Figure 12A,e). After more than 50% crushing of the remaining spine segment, a significant increase in the stress is observable and was characterized as densification phase (Figure 12A,f).

Failure mode II is characterized by an additional, brief period in which the cortex includes its own failure behavior before reaching the maximum compressive strength of the spine segment (Figure 12B). The cortex starts to dismantle slowly from the spine structure after the linear elastic increase, which can be observed, since the coloring of the cortex appears to be lighter in the microphotographs compared to the undamaged cortex (Figure 12C,b). The dismantling of the cortex occurs at 65 MPa being an inflexion point in terms of the stress gradient: the gradient of the stress curve levels off after the dismantling of the cortex. The dismantling balanced the maximized stress acting on the core-cortex interface (Figure 12C,c). The longitudinal spallation of the cortex in large quantities occurs after the maximum compressive strength has been reached. Thus, the core becomes exposed (Figure 12C,d). Horizontal and vertical cracks start to propagate simultaneously through the core within few milliseconds (Figure 12C,e). A significant decrease in the stress can be observed due to the complete crushing of the upper section of the core. Several lath-like segments separate thereafter from the remaining core (Figure 12B,f). A vertical tilting of the lath-like segments occurs with continuing compression; therefore, the stress drops again. The last stage of failure mode II is also characterized by a densification of the remaining segments (Figure 12B,g).

The specific energy absorption per volume unit has to be considered in order to quantify the mechanical effectiveness of the failure modes from the spine segments. The parameter $U_{v}\left(\varepsilon \right)$ is composed of the ratio of the energy absorption, W(ε) and volume of the spine segment, V(ε), destroyed during compression:(2)$$U_{v}\left(\varepsilon \right)=\frac{W\left(\varepsilon \right)}{V\left(\varepsilon \right)}$$

The energy absorption, W(ε), is a measure of the total amount of energy that was dissipated during straining and calculated by integrating the area under the stress-strain curve, which is given as Equation (3):(3)$$W\left(\varepsilon \right)=\int_{0}^{\varepsilon }\sigma \left(\varepsilon \right)d\varepsilon$$

A normalized value of the energy absorption capacity of materials is the energy absorption efficiency, which was adapted by metal foams literature. The energy absorption efficiency, $\eta_{\left(\varepsilon \right)}$, characterizes a materials ability to absorb stress as it compresses by normalizing the energy absorbed by the maximum stress observed [46]: (4)$$\eta_{\left(\varepsilon \right)}=\frac{1}{\sigma_{\max}\varepsilon }\int_{0}^{\varepsilon }\sigma d\varepsilon$$

The peak value of the stress during uniaxial compression is designated as σ_max_. Figure 13A shows the average values of the energy absorption per volume unit at the strain of 0.3, 0.5 and 0.7 calculated by Equation (2). Their normalized average values according to Equation (4), expressed as energy absorption efficiency, are displayed in Figure 13B. The average values of the energy absorption efficiency of failure mode I drops to an average value of 65% at a strain of 0.7 (Figure 13B, black squares). A huge decrease of the average values of the energy absorption efficiency can be determined for the spine segments undergoing failure mode II: the average values of the energy absorption efficiency decrease down to 30% at a strain of 0.7 (Figure 13B, red squares).

Both failure modes are characterized by a quasi-ductility based on multiple fracturing indicating that the spine segments are able to dissipate energy through the generation of a horizontal and vertical crack system and segmentation. Both failure modes differentiate in terms of the intensity of the structural segmentation of the spine segments. During compressive loading, the exposed core (failure mode II) is divided structurally into several lath-like segments. Therefore, the structural integrity is no longer ensured and the capability to absorb energy is, thus, reduced greatly. In contrast to that, the rigid core–cortex connection (failure mode I) is able to reduce the intensity of the structural segmentation. A complete segmentation of the spine segment does not occur in this case. The major proportion of the spine segment remains the same, even though small lath-like segments flake off laterally. Since the quantity of the remaining material is larger compared to the spine segments characterize by failure mode II, the spine segments are, therefore, still able to dissipate energy even at high strain levels. This, in turn, results in comparable large values of the energy absorption per volume unit and energy absorption efficiency for spine segments being subjected to failure mode I. The cortex is able to keep the structural integrity of the spine segments even at advanced stages of compression. 

The failure modes are based on the microstructural configuration of the spine segments. Both failure modes include the mode of segmentation as failure mechanism. Uniaxial experiments of prepared core cylinders (= medulla, radiating layer) reveal the relationship between the strut configuration and the propagation of cracks. For observing the crack propagation in the core cylinder, the uniaxial loading was stopped after the first acoustic signal occurred. Compression experiments on the core cylinder disclose two stages of crack propagation: initial and advanced stage of crack propagation (Figure 14). The first cracks start to propagate directly along the border between the medulla and radiating layer (Figure 14, red box), which is clearly marked as initial stage of the crack propagation. It indicates that the stress is mainly concentrated at the interface between the medulla and radiating layer. Continuing the compressive loading on the core cylinder, the cracks propagate outwards along the stereom sheet of the radiating layer (Figure 14). Therefore, a segment spalls off from the structure and the inner spine microstructure remains unchanged. No cracks appear in the spine’s inner microstructure.

### 3.2. Anisotropic Ceramics

#### 3.2.1. Bioinspired Structural Graded Ceramic

A structural graded ceramic with varying cell sizes on the µm scale is producible using the modified mold design (Figure 3B) and experimental conditions for preparing the suspension described in Section 2.2. The gradation is restricted to the lower region of the mold being directly adjacent to the bottom plate of the mold. Therefore, the distinct structural gradation is present in the first 10% of height from the ceramic. Nevertheless, the fabrication of a graded pore structure with anisotropic cell channels in the ceramic is possible in one single process step. The structural graded ceramic appearing similar to the core–shell structure of the sea urchin spine is displayed in Figure 15A. The inner center of the ceramic comprises polygonal cell channels with almost directional cell channels (parallel to the freezing direction) and is termed as ‘Zone C’ (Figure 15B): the pore diameters vary between 50 and 180 µm (Figure 15C). Further outwards the diameter of the polygonal cell channels becomes smaller. This area is declared as ‘Zone B’. Their pore diameters are between 30 and 50 µm whereby larger pores are visible at few other areas (Figure 15A,C). The cell channels are oriented almost parallel to the freezing direction and tilt increasingly sidewards towards the outside. Very large channel sizes are present in the outermost area of the ceramic, namely in ‘Zone A’ (Figure 15A). Due to the great variation of the cell sizes in ‘Zone A’, it is not feasible to determine an approximate value for the cell sizes in this zone. 

#### 3.2.2. Ceramics with No Gradation

The layered strut arrangement of the radiating layer and the polygonal cell arrangement of the medulla can be transferred in individual ceramics during freeze-casting using conventional cylindrical molds (Figure 3A). The pore morphology is influenced by the composition of the suspension. Both ceramic types (referred here as polygonal and oblate cells, Table 1) have the same solid loading of 14.4 vol.% in common and are, thus, characterized by an average porosity of approximately 79.5%. Different pore morphologies resembling to the spine’s radiating layer and medulla can be manufactured using gelatin as additive in the water-based alumina suspension. Using a gelatin concentration of 3.5 vol.% leads to cellular oblate pore structures (perpendicular to the freezing direction; Figure 16A). The oblate cells are largely aligned in one direction. At some places, comparably smaller cells interrupt the sequence of large oblate cells, but under maintenance of the equal cell orientation in principle. Ceramic bridges are within the large oblate cells. The high anisotropic character of the cell channels can be seen in Figure 16B. Polygonal cellular cell shapes can be manufactured with a gelatin concentration of 6.8 vol.% (Figure 16C). It appears that the directionality of the cell channels becomes vaguer and obtains more a foam-like character (Figure 16D). As can be clearly seen that the configuration of the pore system, i.e., the degree of the interconnectivity of the cell walls is dependent on the gelatin concentration in the suspension.

The results of the maximum compressive strength and Young’s modulus of the freeze-casted alumina ceramics are summarized in Figure 17. Both ceramics are characterized by the same porosity range, but differentiate in their average values for the strength and stiffness (Figure 17A,B). The ceramics comprising polygonal cellular cell channels are characterized by an average compressive strength and Young’s modulus of 1700 MPa and 37 GPa, respectively. Much lower strength and stiffness display the ceramics being characterized by the large cellular oblate cells. This ceramic type has an average value for the maximum compressive strength of 900 MPa. The average Young’s modulus is 25 GPa. 

The stress–strain curves and corresponding microphotographs of the fracture behavior of the freeze-casted ceramics are demonstrated in Figure 18. The ceramic characterized by the polygonal cellular cells compensate the stress by crumbling after the linear elastic increase (Figure 18a,b; white letters). Therefore, the structure flakes off into smaller parts starting from the upper region of the ceramic (Figure 18c,d; white letters). These two fracture mechanisms are capable to avoid the decrease of stress in large scales. By the mode of crumbling and flaking the ceramic is rather capable to keep the acting stress on a constant level. Compared to this, the ceramic with its cellular oblate anisotropic cells is characterized by a steeper gradual decrease of the compensating stress (Figure 18, red curve)). After reaching the maximum compressive strength, vertical cracks propagate through the structure intensifying with progressing compression (Figure 18a–c; red letters). This multiple propagation of vertical cracks results in a division into several lath-like segments tilting increasingly sidewards (Figure 18d,e; red letters). 

## 4. Discussion

### 4.1. Freeze-Casted Ceramic with Gradation Features Inspired by Natural Core–Shell Systems

A modified mold design in the freeze-casting procedure allows to fabricate a structural graded ceramic being similar to concentric core–shell systems found in nature (e.g., plant stems and spine of *Phyllacanthus imperialis*). Only one single processing step is required to achieve different concentric cellular structures distinguishing according to their pore channel sizes and orientations. The differentiation in cellular substructures, anisotropy and pore channel size of the freeze-casted ceramic is similar to the lightweight construction of the spine. A comparable visual display of the spine microstructure (in particular of the interface between the medulla and radiating layer) and structural graded cellular ceramic (Zone B and C) demonstrate that the pore channel sizes are not identical (Figure 19A,B), but are very close together and have, thus, the same order of magnitude. The distribution of the porosity of the structurally graded ceramic corresponds to the cellular stereom structures of the radiating layer and medulla: both are characterized by an ascending porosity from the outside to inside. 

The polygonal cellular shaping and their directionality of the pores from the ceramic-based manufacturing concurs with the shaping and anisotropy of the pores of the sea urchin spine. The structural graded ceramic and the biomimetic concept generator share a lot of common features like the differentiation in several cellular, anisotropic structures and the polygonal cell shape. These characteristics are also present in cellular, organic core–shell systems of plant stems [7,8,47]. 

An implementation of a protective, but permeable external cover such as the lance sea urchin spine’s cortex was only possible on a limited basis. The orientation of the pore channels agrees with the spine’s cortex, but differentiate in terms of the wide distribution of the pore channel sizes. The cortex is characterized by a highly branched pore channel system with an average constant pore channel size of 17 µm. In contrast, the pore channels in the cortex-alike structure in the ceramic can obtain dimensions of >200 µm in diameter appearing as dense pore channel bundle in the microstructure. As a consequence, the quantity of solidified material between the cell channels is relatively small compared to the spine’s cortex. Microstructural observations (see Figure 15, Zone A) have demonstrated that the almost dense cortex structures of the spine cannot be translated properly into a ceramic structure. The detailed wedge-like structures of the spine (i.e., the superstructure) cannot be transferred either. The fact that the cortex structure and superstructures of spine could not be transferred into a ceramic structure in their fully extent is due to the manufacturing technique itself. The subtleties of the stereom layers as well as their wedge-like enclosure of the lateral branches of the medulla cannot be depicted in detail with freeze-casting. Therefore, the superstructure has to be simplified within the framework of the possibilities of the freeze-casting method: the radial character of the pores and accompanied cell walls can be improved by lowering significantly the freezing temperatures using the mold design in Figure 3B. Significantly lower freezing temperatures enhance the ice crystal growth and its radial propagation through the suspension. The increased propagation velocity of the ice crystals allows to disrupt the discontinuous gelatin–water network more easily. Radially ordered cellular structures could be obtained in this way resembling the radially arranged stereom sheets in the sea urchin spine’s radiating layer and the intermediate layers of plant stems in a simplified manner. With this processing procedure, differentiated structures similar to the cellular microstructure of the sea urchin spine’s core (medulla and radiating layer) and of plant stems can be fairly transferred into a structurally graded ceramic. This one-step procedure, however, excludes the manufacturing of a similar ceramic-based structure to the spine’s cortex. Since a suspension with a particular solid content is frozen, substructures can be fabricated in the ceramic that differ only minimally in porosity. The radiating layer and the medulla are characterized by a porosity of 76.6 ± 3.0 and 85.7 ± 2.8%, respectively. The difference in the porosities between them is comparatively small and is realizable in one single ceramic via freeze-casting. In contrast, the difference in the porosities between the elements of the spine’s core and the cortex is large. In order to realize an external cover such as the spine’s cortex in the ceramic, a second radial freezing step is required involving a suspension with a higher solid content.

### 4.2. Freeze-Casted Ceramics with Anisotropic Cellular Structures

The manufacturing of a porous ceramic with various gradation features is essentially dependent on the mold design in the freeze-casting process. Fabrication of individual anisotropic cellular structures in the ceramic involving no gradation features at all is rather a function of the composition of the suspension and freezing condition. By the use of gelatin as additive in the water-based alumina suspension, cellular pores systems on the µm scale can be manufactured resembling structures found in nature. Depending on the gelatin concentration in the suspension anisotropic polygonal channels (see Table 1) and cellular oblate pore systems (see Table 1) are producible that appear very similar to the spine’s interior, i.e., the medulla and radiating layer, respectively. These ceramic structures, in particular the anisotropic polygonal pore systems look also similar to other cellular materials in plants such as the cross section of a milkweed stem [48] and a porcupine quill [7]. Freeze-casting is, therefore, a practicable manufacturing technique to create a multitude of anisotropic plant-alike cell structures in ceramics. 

The microcellular ceramics shown here differ in terms of their strength and stiffness, even though they have similar porosities. Therefore, the strength and stiffness of the ceramics strongly relate to the shaping of the pore system. The cross sections in Figure 20 (2D view, perpendicular to the freezing direction: complete filling of the cavities with epoxy resin) reveal that the cell walls are comparatively stronger interconnected due to the polygonal pore shape in the ceramic. Larger stiffness and strength are a result of the high interconnectivity of the cell walls (Figure 20A). In contrast, several cell walls are isolated in space and, thus, unconnected in the cellular oblate cell system (Figure 20B). Hence, comparatively lower values in the strength and stiffness are the result of the reduced interconnectivity in the oblate cell system. The fracture behavior of the ceramics under uniaxial compression is likewise strongly related to the shaping of the cell system: a progressive interplay of crumbling and flaking of small lateral pieces was observed for the ceramics with the polygonal anisotropic cell channels; the segmental division into several lath-like segments caused by a multitude of vertical cracks was a characteristic for the ceramics comprising the oblate cellular cell system. Since several cell walls are isolated in space in this latter ceramic type, the cell walls are subjected to high shear stresses and bending moments and, therefore, susceptible to high deflection during uniaxial compression. Consequently, large segments can spall off from the structure. The structural integrity is no longer ensured and the capability to absorb stress is, thus, reduced greatly. The greatly linked anisotropic polygonal cell architecture prevents shearing and bending moments of the cell walls. The stiffing, therefore, increases the mechanical stability and keeps the material loss comparatively small. It implies that the major quantity of the ceramic tends to retain its structural integrity. Consequently, a high quantity of stress is, therefore, required to break the strong cohesive structure. Both main fracture modes in the ceramics have the structural segmentation in common. They differentiate in terms of the intensity of the structural segmentation or destruction being comparable to the mode of ‘quasi brittle’ fracturing of the sea urchin spine. 

There is a structural difference between microstructure of the sea urchin spine and the cellular ceramics presented here. The key element of the structural segmentation for the freeze-casted ceramics is dependent on the degree of interconnectivity of the cell walls. In contrast, the structural segmentation or destruction of the sea urchin spine is dependent on the core’s wedge-shaped interlocking of the microstructure and of the quantity of the surrounding cortex.

### 4.3. The Plant Stem-Alike Core–Shell Construction in the Lance Sea Urchin Spines

The core of the spine with its cellular substructures (i.e., the medulla and radiating layer) is characterized by a highly interconnected complex stereom network: two different stereom types are united in a special arranged wedge-shaped configuration according to the key–lock principle (i.e., a superstructure). The intelligent wedge-alike interlocking saves building material and deflect cracks advantageously. Uniaxial compression experiments on the core have demonstrated that the stress is concentrated at the interface between the medulla and radiating layer. Studies on natural core–shell systems of plant stems, animal quills and bird feather rachis under uniaxial compression [7,44,45] have shown that the stress is also maximized at the interface and decays radially inward. So, the first cracks appear at the interface and run along the stereom layers to the exterior of the core. Therefore, lath-like segments emerge in the spine structure. The size of the separated segments could be limited by the wedges in principle to keep the material loss as low as possible. With the minimal use of building material, the core of the spine is optimized in terms of the ability to provide stiffness, strength and an advantageous energy dissipation. In addition, the inner complex mesh network is surrounded by the permeable shell structure—the cortex. 

The quantity of the cortex in the spine segment dictates in the main the dimension of the structural integrity and the resilience of the spine segment to mechanical stress. A high-volume fraction of the cortex at the spine tip (a/t ratio > 9.2), which decreases towards to the spine base (a/t ratio < 9.2), may indicate a possible adaption of the sea urchin to its natural habitat. The sea urchin wedges themselves during the day time in the reef cavity in order to protect themselves against the high hydrodynamic environment and predators [49,50]. The spine tip is directly in contact with the hard substrate due to the wedging. A high resilience at the spine tip is, therefore, required, since a high punctual load, which lasts for hours, act on the tip of the spine. The high-volume fraction of the cortex at the spine tip is, thus, useful to enhance the resilience in principle. The ‘quasi bulging’ of the cortex (failure mode II) might be an indication for a reduced rigidity. A solid solution hardening of the material is achieved by the magnesium incorporation in the crystal structure [51]. Investigations have shown that the average magnesium concentration in the cortex is 5.1 mole percent MgCO_3_ [52]. This is significantly lower than the concentrations observed in the medulla (5.9 mole percent MgCO_3_) and the radiating layer (8.1–0.5 mole percent MgCO_3_). A decrease of the magnesium in the cortex reduces also the degree of brittleness, which allows, therefore, minimal elastic reactions. These minimal elastic reactions are expressed in pure compressive loading of the spine segments as ‘quasi bulging’. A minimal degree of elasticity of the cortex at the spine tip is required, since the sea urchin has to modify the pressure for the wedging according to the prevailing hydrodynamic conditions. Changing flow velocities and the occurrence of predators imply a highly dynamic environment for the sea urchin. The pressure acting on the spine is constantly readjusted to the changes of the environment. Therefore, the punctual load at the spine tip fluctuates during the wedging. A rigid cortex would directly lead to the actual material fracture of the upper spine segment. This interpretation of the results extends the rather chemical function of the cortex given by [52]. They have argued that the low magnesium concentration in the cortex compared to the inner core and the increased density of the cortex is responsible for the large acidification resistance. Probably the large acidification resistance enabled them to survive the Permian-Triassic crisis, which was associated with severe acidification events of the ocean. Segments from the lower half of the spine are characterized by a rigid cortex–core connection. Therefore, these spine segments keep widely the structural integrity under high compressive loading and maintain the degree of segmentation as low as possible. Conversely, segments from the spine tip are strongly subjected to structural segmentation, since the cortex was spalled off right at the beginning of the experiment. Therefore, the core was not stabilized anymore by the cortex. The lower half of the spine has been optimized towards keeping further material losses as low as possible, when the upper half of the spine does not exist anymore.

The level of compressive loading acting on the sea urchin spine segment in the experimental setting (>500 N) was significantly higher than the mechanical loading conditions prevailing in the natural habitat of the sea urchin. Such compressive loading conditions in the experimental setting would probably press the spine into the test. The test would fracture, thus, catastrophically before the spine would dissipate energy. Therefore, the ability to dissipate energy advantageously is not necessarily an adaption to its mechanical environment, but rather a positive by-product of the porous, hierarchically-organized material by itself. Nevertheless, the results of the mechanical testing of the spine segment offered room to discuss the function of the structural elements of the spine in principle.

## 5. Conclusions

The manufacturing of a porous ceramic with several gradation features inspired by natural cylindrical core–shell systems is possible using freeze-casting as processing technique. Using a special mold design and optimized suspension for the freeze-casting technique allows to achieve a differentiation in several cellular substructures, anisotropy and pore channel sizes in the ceramic comparable to the microstructure of plant stems and the spine of *Phyllacanthus imperialis*. Biological structures such as the permeable cortex of spine and the regular arrangement of the stereom sheets (=radiating layer) were only transferred into a ceramic structure to a limited degree. Several substructures can be fabricated in one single ceramic that differ only minimally in the porosity, since a suspension with a particular solid content is used for the freezing process. Therefore, structures such as the spine’s medulla and radiating layer are manufacturable in an abstracted way in ceramics using one single processing step, because the difference in the porosities between them is very small (approximately 10%). However, for realizing a cortex-alike structure in the ceramic, a second radial freezing step is required involving a suspension with a higher solid content. 

Ceramics with individual cellular pore systems (i.e., no gradation features) are possible using a conventional mold design during the freeze-casting process. Ceramic structures being similar to cellular materials in plants such as milkweed and porcupine stem can be manufactured using gelatin as additive in the water-based suspension. Starting at a gelatin concentration of 6.8 vol.% with a gradual reduction of the gelatin concentration by 50% has shown that polygonal and cellular oblate pore systems can be manufactured in this way. A reduced lamellar spacing through the use of gelatin is an important structural factor that contributes to the improvement of mechanical properties. The structural modification through gelatin brings the overall structure closer to that of a honeycomb using its improved mechanical efficiency. The increased number of stiffening and strengthening bridges stabilizes the overall structure through rib stiffening and reduce failure due to shearing. 

The new results focusing on the relation between the material properties and the specific microstructural configuration of the spine offer room for discussion in the light of the mechanical environment of the sea urchin *Phyllacanthus imperialis*. The wedge-shaped interlocking of two different stereom meshes according to the key–lock principle of the spine’s inner microstructure is a biological optimized structure to provide stiffness, strength and a beneficial energy dissipation with the minimal use of material. The surrounding permeable shell structure, the cortex, dictates the dimension of the structural integrity and the resilience of the spine to mechanical stress. A high-volume fraction of the cortex at the spine tip enhances the mechanical stability and is required, since the sea urchins wedge themselves between the reef cavities during the day time. A high resilience at the spine tip is, therefore, required to withstand the high punctual loadings, which can last for hours. The cortex is characterized by lower magnesium concentrations compared to the inner structural elements of the spine. Low magnesium concentrations reduce the rigidity of the cortex and allow minimal elastic reaction that was experimentally visible under extreme loading conditions of the spine segment (‘quasi bulging’). A minimal degree of elasticity of the cortex at the spine tip is reasonable, since the sea urchin have to modify the pressure for the wedging according to the prevailing hydrodynamic conditions. In contrast, segments from the lower half of the spine are characterized by a rigid cortex-core connection. Thus, the lower half of the spine has been optimized towards minimizing further material losses, when the upper half of the spine has been broken off. 

Porous anisotropic ceramics are widely used as catalyst supports and as particulate filters for vehicular emission control. New geometries and materials are permanently developed for both mobile and stationary applications. As a result, considerable work is done studying higher cell density and thinner wall configurations. A higher cell density and thinner cell walls of the substrates increase the geometric surface area and reduce the hydrocarbon emissions during all phases of the catalytic conversion. To avoid a decreasing substrate strength due to thinner ceramic walls, the special key–lock arrangement of the wedges and the permeable shell-like construction from the lance sea urchin spine can be integrated in the porous filter system. Hereby, a mechanical strengthening of the overall structure is possible while maintaining the necessary flow properties due to the anisotropic pore morphologies. 

## Figures and Tables

**Figure 1 biomimetics-06-00036-f001:**
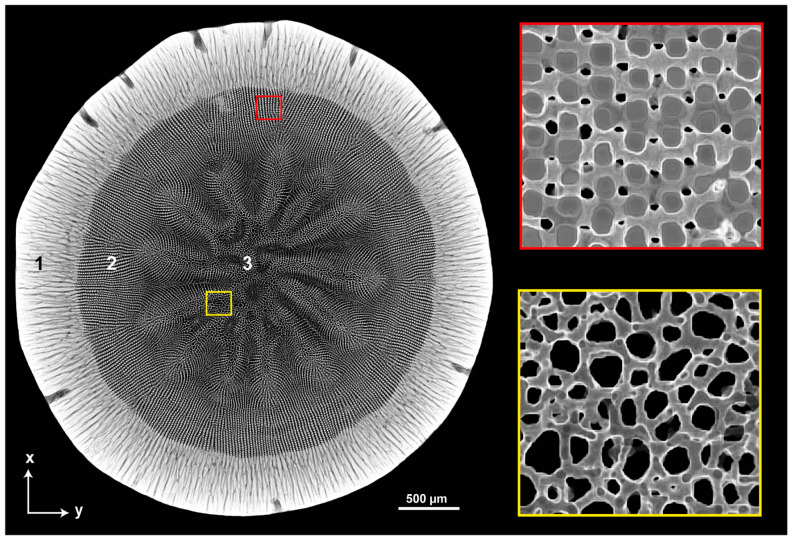
Cross sectional image of the aboral spine of *Phyllacanthus imperialis* based on µCT data visualizing the core–shell structure with its differentiated substructures. 1: Cortex, 2: Radiating layer, 3: Medulla.

**Figure 2 biomimetics-06-00036-f002:**
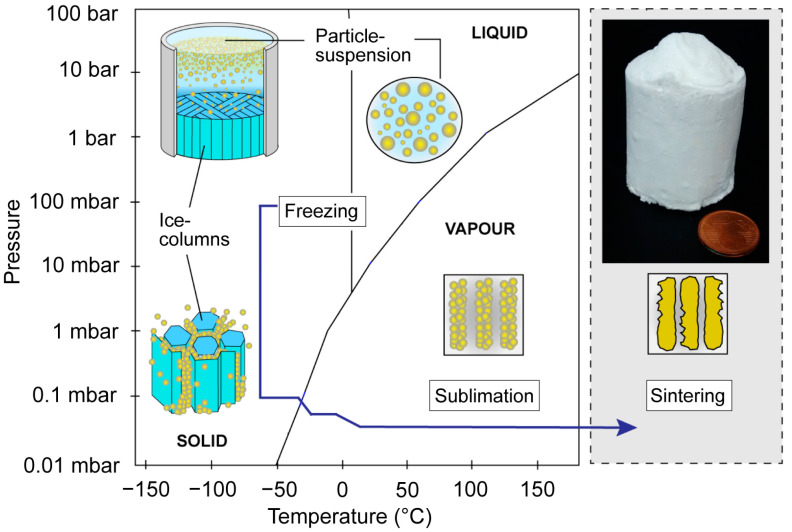
Schematic overview of the freeze-casting process with four main processing steps: preparation of the suspension, solidification via freezing, sublimation and sintering.

**Figure 3 biomimetics-06-00036-f003:**
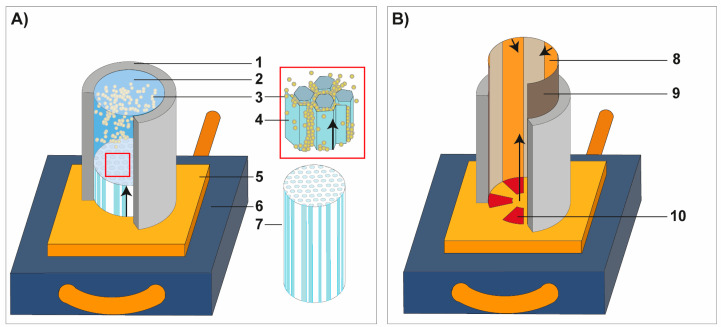
Schematic illustration of the mold designs for the freeze-casting process. (**A**) The conventional setup for producing unidirectional pore systems consists of a polytetrafluorethylene (PTFE) tube that was fixed on a copper plate. The mold design in (**B**) was used to manufacture a structural graded ceramic inspired by the microstructure of spine of *Phyllacanthus imperialis*. The freezing direction is indicated by the black arrows. 1: PTFE-mold, 2: particle suspension, 3: Al_2_O_3_ particle, 4: ice crystal, 5: copper plate, 6: cold plate, 7: frozen suspension with unidirectional ice crystals, 8: copper film, 9: Teflon film, 10: silicon film.

**Figure 4 biomimetics-06-00036-f004:**
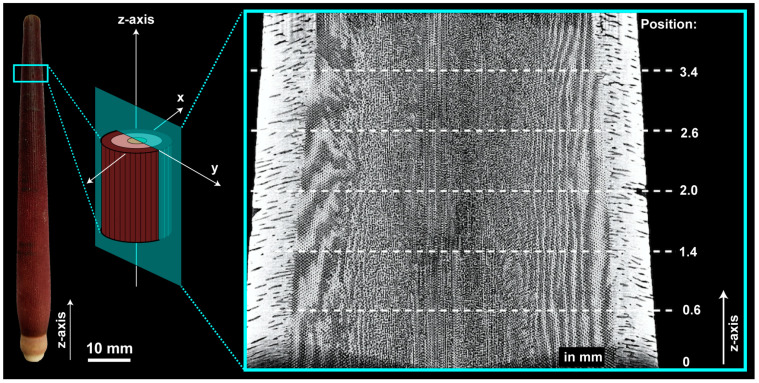
Spine of *Phyllacanthus imperialis*. Turquoise box illustrates the scanned area via μCT. A complete reconstruction of the scanned spine area can be seen on the right-hand side.

**Figure 5 biomimetics-06-00036-f005:**
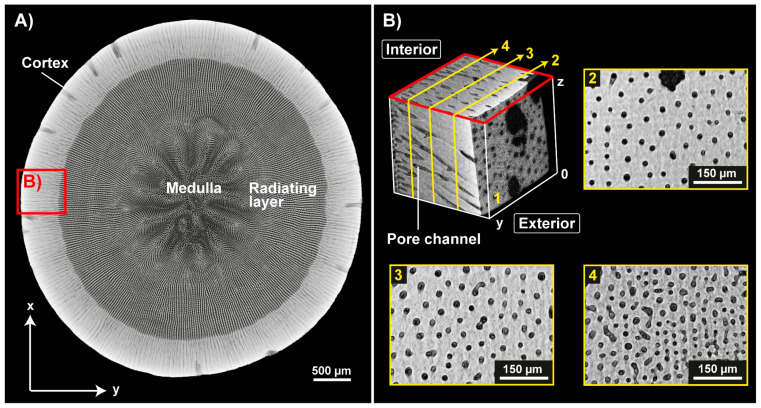
Microstructure of the spine of *Phyllacanthus imperialis*. (**A**) Z-projection of the microstructure of the spine shows the three structural units: the cortex, radiating layer and medulla. Various sections (yellow boxes) of the cortex microstructure are displayed in (**B**).

**Figure 6 biomimetics-06-00036-f006:**
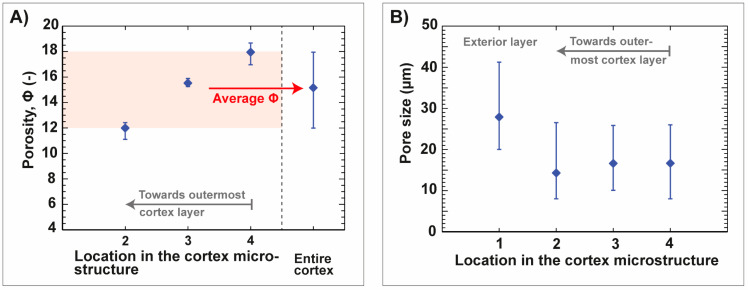
Porosity, Φ, of the cortex and the size of the pore channels within the cortex of the spine of *Phyllacanthus imperialis*. The porosity of the µCT sections of the cortex is displayed in (**A**). The average porosity of each μCT section is displayed in (**B**). The assignment of the µCT sections (2, 3, 4) is displayed in Figure 5B.

**Figure 7 biomimetics-06-00036-f007:**
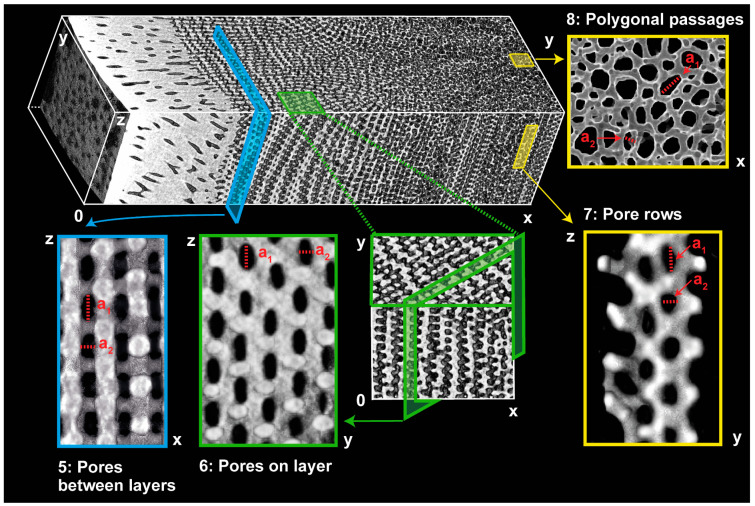
A stacked µCT reconstruction of the inner microstructure of the spine of *Phyllacanthus imperialis*. The turquoise and green boxes visualize different cross-sectional planes of the structure of the radiating layer. The inner mesh of the center of the spine is shown in the yellow boxes (polygonal passages, pore rows). The largest and the smallest pore diameter are indicated by a_1_ and a_2_, respectively.

**Figure 8 biomimetics-06-00036-f008:**
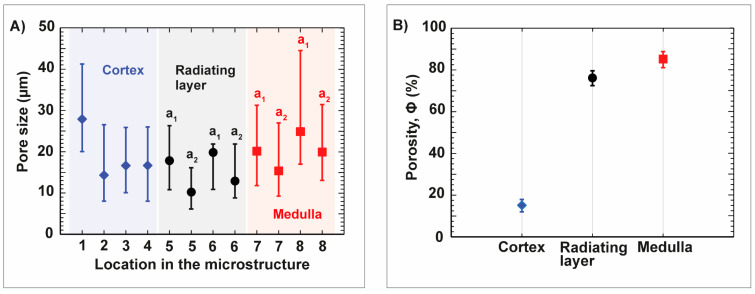
Characterization of the pore sizes and porosities of the microstructural units (cortex, radiating layer and medulla) in the spine of *Phyllacanthus imperialis*. The pore axis length is shown by a_1_ and a_2_, and of the structures forming the radiating layer, the medulla and the cortex are displayed in (**A**). The location within the spine microstructure is given as numbers. The affiliation of the numbers is shown in Figure 5B and Figure 7, respectively. The plot in (**B**) represents the average porosities of the cortex, radiating layer and medulla.

**Figure 9 biomimetics-06-00036-f009:**
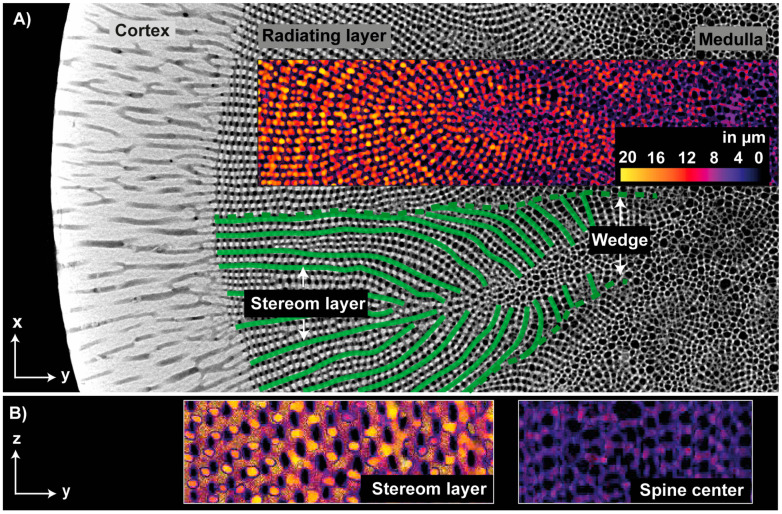
(**A**) µCT reconstruction of the spine of *Phyllacanthus imperialis* demonstrating the calculated strut thickness distribution of the meshes (stereom layers and polygonal meshes in the spine center) and a superstructure. The stereom layers (marked in green) running directly from the cortex to one sidearm of the spine center (xy-plane). Thus, a wedge-like structure (dashed line in green) is formed, which is a superstructure. (**B**) Cross sections, parallel to the *z*-axis, of the stereom layer and spine center. The color scheme in (**A**,**B**) indicates the strut thickness.

**Figure 10 biomimetics-06-00036-f010:**
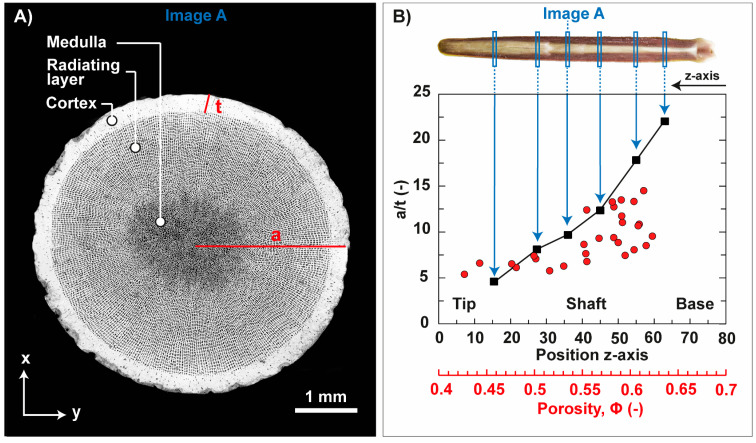
Quantity of the cortex in the microstructure of the spine of *Phyllacanthus imperialis*. (**A**) The quantity of the cortex is expressed as a/t ratio. The a/t ratio is composed of the radius of the spine segment, a, which was normalized by the cortex thickness, t. The quantity of the a/t ratio and the porosity at specific positions in the spine is demonstrated in (**B**) as black squares and red circles, respectively.

**Figure 11 biomimetics-06-00036-f011:**
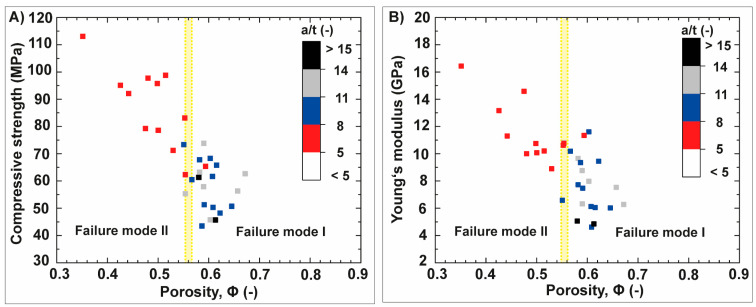
Plots in (**A**,**B**) show the maximum compressive strength and Young’s modulus of the spine segments of *Phyllacanthus imperialis* as a function of the porosity, Φ. The color bar indicates the structural factor of the spine segments, the a/t ratio. A change of the failure modes was observed at a porosity of approximately 55% being highlighted in the plots as light-yellow line.

**Figure 12 biomimetics-06-00036-f012:**
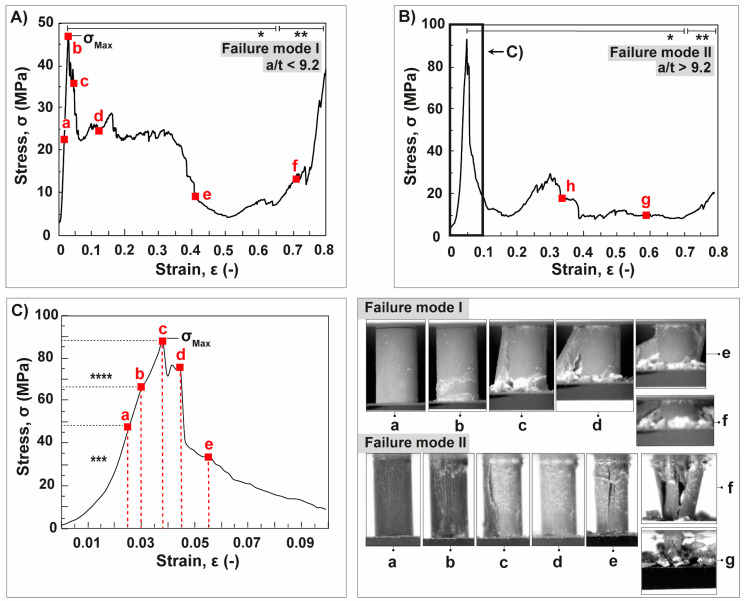
Spine segments of *Phyllacanthus imperialis* under uniaxial compression. Typical stress (σ)-strain (ε) curves of failure mode I and II are demonstrated in (**A**,**B**), respectively. A detailed view of the initial stress–strain curve of a spine segment undergoing failure mode II is displayed in (**C**). The red letters belong to the microphotographs presenting the fracture behavior at a specific stage of uniaxial compression. *: ‘Quasi-ductile’ regime. **: Densification. ***: Linear elastic regime with a brief alignment period. ****: Dismantling of the cortex.

**Figure 13 biomimetics-06-00036-f013:**
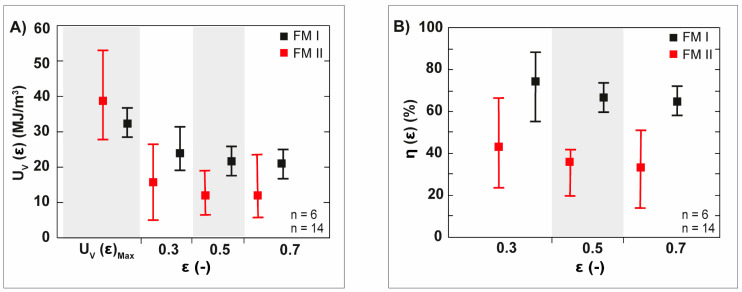
Specific energy absorption per volume unit, U_V_(ε), and the energy efficiency, η(ε), of the spine segments of *Phyllacanthus imperialis*. The relation of the average values of the U_V_(ε) and η(ε) to the strain, ε, is displayed in (**A**,**B**), respectively. FM I: Failure mode I, FM II: Failure mode II.

**Figure 14 biomimetics-06-00036-f014:**
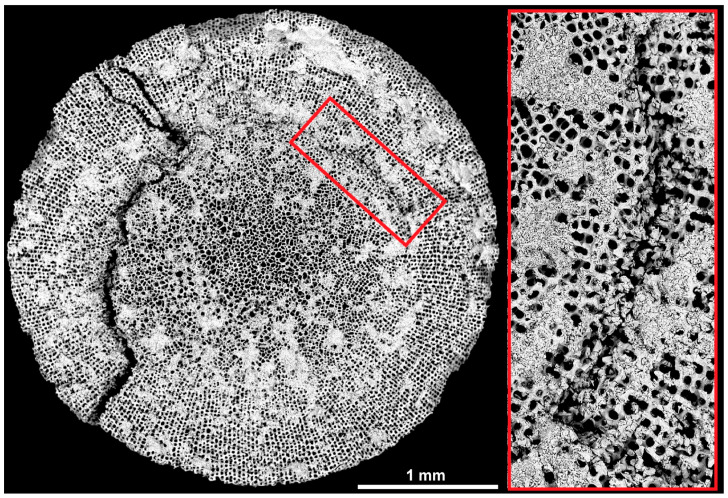
Cross sectional BSE image of the spine’s core of *Phyllacanthus imperialis*, which was uniaxial loaded until the first acoustic signal occurred. Initial crack propagation at the medulla-radiating layer interface is displayed in high magnification in the red box. The advanced stage of crack propagation is demonstrated on the left-hand side: the crack is deflected along the stereom sheet.

**Figure 15 biomimetics-06-00036-f015:**
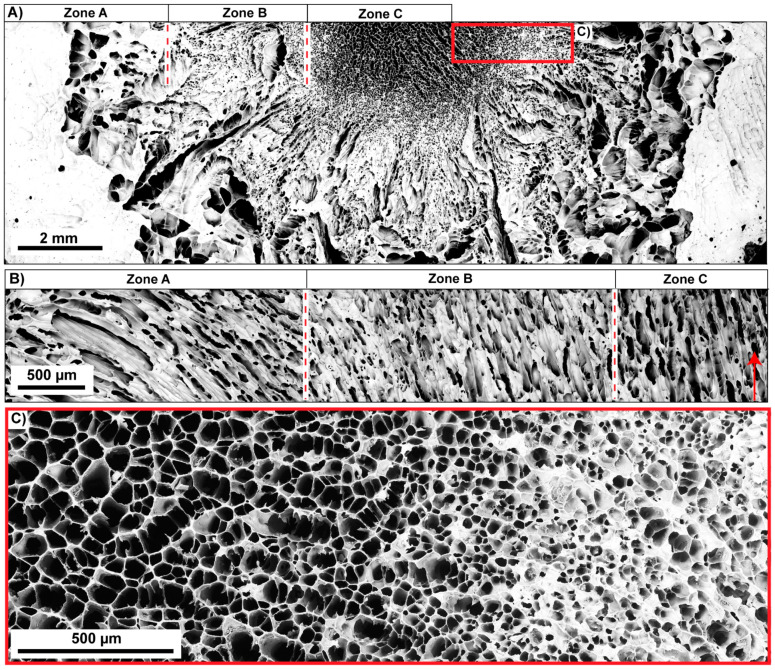
Structural graded ceramic inspired by the microstructure of *Phyllacanthus imperialis*. The microstructure of the ceramic and its division in different zones is displayed in (**A**) that illustrates the cross-sectional area perpendicular to the freezing direction. The orientation of the cell channels is given in (**B**). The red arrow demonstrates the freezing direction. A higher magnification of the graded structure is displayed in (**C**). The area is marked as red rectangle in (**A**).

**Figure 16 biomimetics-06-00036-f016:**
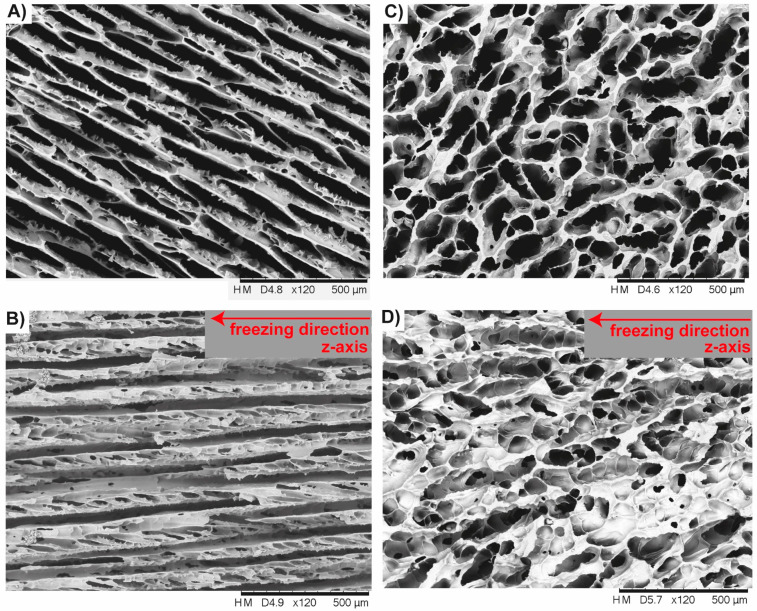
Freeze-casted cellular ceramics inspired by the strut arrangement of the spine’s radiating layer and medulla. Both ceramic types have the solid loading of 14.4 vol.% in common and differentiate in terms of the used gelatin concentration in the suspension. A gelatin concentration of 3.5 vol.% has been used to manufacture the cellular oblate cell structure, (**A**), being highly anisotropic, (**B**), in their pore shape. Polygonal cellular cell shapes, (**C**), can be manufactured with a gelatin concentration of 6.8 vol.% showing rather a foam-like character, (**D**).

**Figure 17 biomimetics-06-00036-f017:**
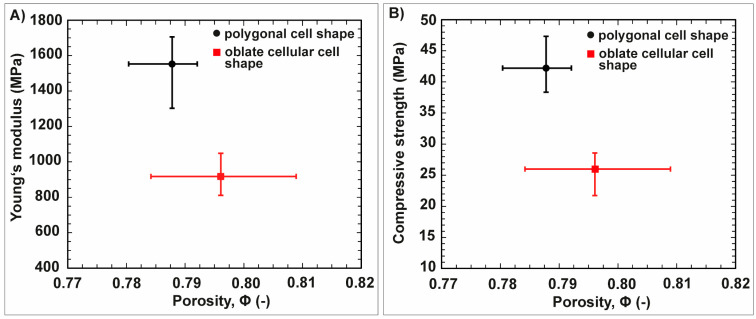
The plots in (**A**,**B**) show the average Young’s modulus and maximum compressive strength of the freeze-casted ceramics as a function of the porosity, Φ.

**Figure 18 biomimetics-06-00036-f018:**
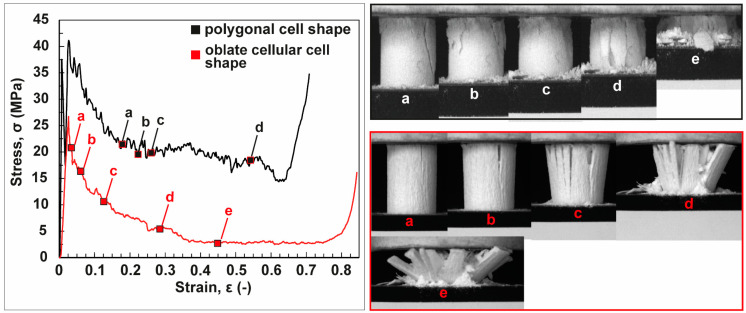
The stress (σ)-strain (ε) curves and the corresponding microphotographs of the fracture behavior of the freeze-casted ceramics. The ceramics including the polygonal cells compensate stress by a progressive interplay of crumbling and flaking (a–e, white letters). An intensive lath-like segmentation (a–e, red letters) can be observed for the ceramics being characterized by cellular oblate anisotropic cells.

**Figure 19 biomimetics-06-00036-f019:**
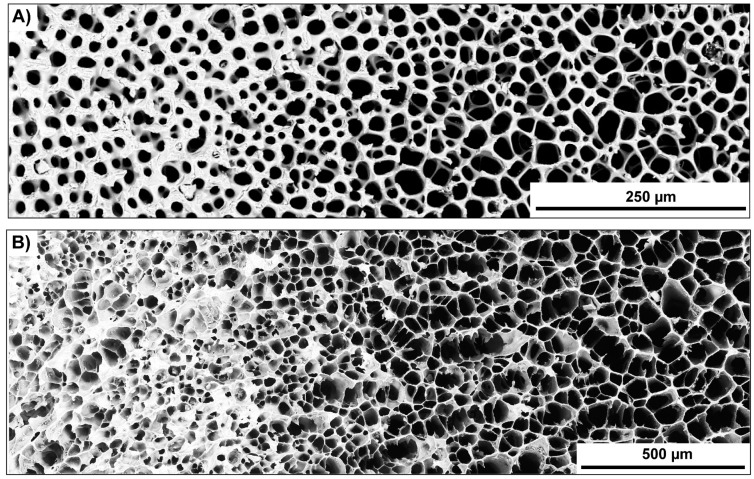
Comparable display of BSE images of the spine microstructure of *Phyllacanthus imperialis* and of the structural graded ceramic manufactured via freeze-casting. The cross section (perpendicular to the *z*-axis) of the spine microstructure is displayed in (**A**) demonstrating the radiating layer and medulla. Part of the structural graded freeze-cast ceramic with its cross section is presented in (**B**).

**Figure 20 biomimetics-06-00036-f020:**
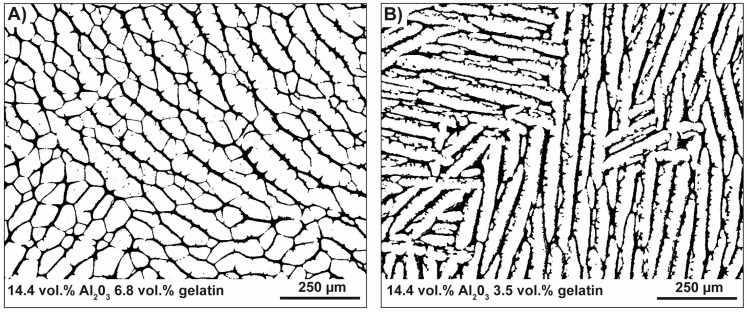
BSE images of the cross sections of the freeze-casted ceramics (perpendicular to the freezing-direction). The cavities of the ceramics were filled with epoxy resin (=white areas). Therefore, the varying degrees of interconnectivity of the struts in the ceramics is demonstrated in (**A**,**B**) depending on the gelatin concentration.

**Table 1 biomimetics-06-00036-t001:** Concentration of the solid loading of Al_2_O_3_ and gelatin of the water-based alumina suspensions.

Name	ϕ Sintered Foam	Total Concentration of Al_2_O_3_		Total Concentration of Gelatin with Regard to Water		Dolapix CE 64
	(vol.%)	(wt.%)	(vol.%)	(wt.%)	(vol.%)	(cm^3^)
Polygonal cells	77–79	44.0	16.5	5	6.8	0.47
Oblate cells	77–79	44.0	16.5	2.5	3.5	0.47
Graded ceramic	77–79	44.0	16.5	5	6.8	0.47

## Data Availability

Data is contained within the article.

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
