# Peer review of "The Plant-Like Structure of Lance Sea Urchin Spines as Biomimetic Concept Generator for Freeze-Casted Structural Graded Ceramics"

_biomimetics, 2021, doi:10.3390/biomimetics6020036_

Round 1
Reviewer 1 Report
The article is well written with interesting insides. There needs to be a better discussion on the practical application of the research - even in a form of speculation. Some parts could have been better illustrated and explained. I attached the paper with some suggestions.

Author Response
Dear reviewer,
We would like to thank the reviewers for their comments. This helped us to improve the quality of the paper. We have thoroughly checked the reviewers’ comments concerning our work and have accordingly made the following modifications and corrections.
Point 1: Please illustrate the manufacturing process of a porous ceramic via freeze-casting.
Response 1: The manufacturing process of a porous ceramic via freeze-casting was illustrated in Figure 2. Figure 2 was added in the manuscript to show the schematic overview of the freeze-casting process.
Point 2: Explain ranges of the process parameters like the slurry concentrations, freezing temperatures and cooling rates.
Response 2: The following text was added to explain the ranges of the process parameters:
It is important that the prepared slurries are stable during the entire duration of the freezing stage. Since the solvent initially present in the slurry is converted into solid, that is later eliminated to form the porosity in the ceramic, the pore content can be adjusted by tuning the slurry concentrations. The porosity of the ceramic is directly related to the volume of the solvent. A wide range of porosities, approximately from 25 to 90 %, can be achieved via freeze-casting [25]. The total porosity is also depended on numerous additional parameters affecting the packing of particles between the solvent crystals such as the nature of the solvent, its viscosity, the particle morphologies and size distribution [26]. The solidification behaviour of the freezing vehicles and thus the pore structure left by the frozen vehicles is affected by the freezing temperatures [25,27–29]. The pore channel size decreases significantly with lower freezing temperatures, regardless of any possible microstructure variations in the individual specimens. Porosity also decreases as freezing temperatures decline. With decreasing freezing temperatures, the solidification velocity is increased inducing smaller lamellar ice crystal spacing and thus pore channel size.
Point 3: Explain what is the environmental impact of the addition of gelatine.
Response 3: The following text was added to explain the environmental impact of the addition of gelatine:
A reduced lamellar spacing through the use of gelatine is an important structural factor that contributes to the improvement of mechanical properties. The structural modification through gelatine brings the overall structure closer to that of a honeycomb using its improved mechanical efficiency. The increased number of stiffening and strengthening bridges stabilizes the overall structure through rib stiffening and reduce failure due to shearing.
Point 4: Please, elaborate on the practical application.
Response 4: The following text was added to underline the practical application of the new findings:
Porous anisotropic ceramics are widely used as catalyst supports and as particulate filters for vehicular emission control. New geometries and materials are permanently developed for both mobile and stationary applications. As a result, considerable work is done studying higher cell density and thinner wall configurations. A higher cell density and thinner cell walls of the substrates increase the geometric surface area and reduce the hydrocarbon emissions during all phases of the catalytic conversion. To avoid a decreasing substrate strength due to thinner ceramic walls, the special key-lock arrangement of the wedges and the permeable shell-like construction from the lance sea urchin spine can be integrated in the porous filter system. Hereby, a mechanical strengthening of the overall structure is possible while maintaining the necessary flow properties due to the anisotropic pore morphologies.

Reviewer 2 Report
Authors have presented detailed analysis and studies the topic extensively. I would recommend the publication of the manuscript without any changes. Author can check the typographical errors and some English sentences.
Author Response
Dear reviewer,
We would like to thank you for reviewing our paper. We have thoroughly checked the latest version of our paper. Attached you can find an improved version of the paper. Some paragraphs contain now more details and have been rewritten to improve the quality of the paper.
